# Transparency of research practices in cardiovascular literature

Gabriel O Heckerman[1,2†], Eileen Tzng[2,3†], Arely Campos-Melendez[2,4†], Chisomaga Ekwueme[2,5], Adrienne Mueller[2]*

[1]Western Kentucky University, Bowling Green, United States; [2]Stanford Cardiovascular Institute, Stanford, United States; [3]Cornell University, Ithaca, United States; [4]University of California, Los Angeles, Los Angeles, United States; [5]University of California, Davis, Davis, United States

## Abstract

**Background:** Several fields have described low reproducibility of scientific research and poor accessibility in research reporting practices. Although previous reports have investigated accessible reporting practices that lead to reproducible research in other fields, to date, no study has explored the extent of accessible and reproducible research practices in cardiovascular science literature.

**Methods:** To study accessibility and reproducibility in cardiovascular research reporting, we screened 639 randomly selected articles published in 2019 in three top cardiovascular science publications: Circulation, the European Heart Journal, and the Journal of the American College of Cardiology (JACC). Of those 639 articles, 393 were empirical research articles. We screened each paper for accessible and reproducible research practices using a set of accessibility criteria including protocol, materials, data, and analysis script availability, as well as accessibility of the publication itself. We also quantified the consistency of open research practices within and across cardiovascular study types and journal formats.

**Results:** We identified that fewer than 2% of cardiovascular research publications provide sufficient resources (materials, methods, data, and analysis scripts) to fully reproduce their studies. Of the 639 articles screened, 393 were empirical research studies for which reproducibility could be assessed using our protocol, as opposed to commentaries or reviews. After calculating an accessibility score as a measure of the extent to which an article makes its resources available, we also showed that the level of accessibility varies across study types with a score of 0.08 for case studies or case series and 0.39 for clinical trials (p = 5.500E−5) and across journals (0.19 through 0.34, p = 1.230E−2). We further showed that there are significant differences in which study types share which resources.

**Conclusions:** Although the degree to which reproducible reporting practices are present in publications varies significantly across journals and study types, current cardiovascular science reports frequently do not provide sufficient materials, protocols, data, or analysis information to reproduce a study. In the future, having higher standards of accessibility mandated by either journals or funding bodies will help increase the reproducibility of cardiovascular research.

**Funding:** Authors Gabriel Heckerman, Arely Campos-Melendez, and Chisomaga Ekwueme were supported by an NIH R25 grant from the National Heart Lung and Blood Institute (R25HL147666). Eileen Tzng was supported by an AHA Institutional Training Award fellowship (18UFEL33960207).

*For correspondence:
alm04@stanford.edu

†These authors contributed equally to this work

Competing interest: The authors declare that no competing interests exist.

## Editor's evaluation

This paper in the field of metascience reports important findings on the levels of accessibility and reproducible research practices in the field of cardiovascular science. As such, it provides a solid benchmarks against which future work could be assessed. The article is of broad interest to basic and clinical cardiovascular scientists.

**eLife digest** Scientists are under pressure to publish impactful research quickly. In this "publish or perish" culture of science, scientists who are the first to publish a discovery in a well-known journal are often rewarded with credit and more funding. But this publication pressure can have unintended consequences and lead to the publication of incomplete or flawed research. To improve the quality and integrity of research, advocates encourage scientists to share their data and how they came to their conclusions to allow others to verify or replicate their work.

Several fields, including cancer research, have launched efforts to improve study quality and transparency. But so far, there has been little analysis about the reproducibility and replicability of cardiovascular disease research. Assessing how much information cardiovascular researchers provide about their methods, materials, and data could help determine the quality of heart disease research studies and boost confidence in the field's discoveries.

Heckerman et al. found that fewer than 2 percent of heart disease research studies include enough information to be verified by other scientists. In the analysis, Heckerman et al. analyzed 393 cardiovascular disease research studies to determine if the studies provided enough information for others to replicate their work. The amount of information the authors shared varied according to study type. Studies describing the experiences of individuals or small numbers of patients shared the least information about how they came to their conclusions, while larger clinical trials shared more. In some cases, such as when a study used personally identifiable patient information or a drug company's proprietary data, there may have been reasons to keep the data confidential.

The findings provide valuable information for both cardiovascular scientists and the public. They show a lot of room to improve trust in cardiovascular disease research by ensuring more studies are verifiable. Scientific journals and research funders could incentivize researchers to share more information about their methods and data to increase trust and transparency in the field.

## Introduction

Previous studies have reported on the lack of detailed methodology, data, and code provided that would be necessary for individual studies or study sets to be considered replicable and reproducible (*Filazzola and Cahill, 2021*). In addition, scientific publications are often behind paywalls that only users affiliated with a larger institution are readily able to access, meaning that research funded by public money is often inaccessible to the general public without paying a fee. Scientific practices have been evolving over time: data from a study can be complex and require a large amount of storage space, methods can be extremely sophisticated and require specialized equipment, and data analysis can require complex algorithms and code. Journals often do not specify requirements for materials sharing, data sharing, analysis code sharing, and methodological information – all of which have been shown to be important to be able to reproduce or replicate a study (*Hamra et al., 2019*; *Munafò et al., 2017*). Note that often there may be legitimate reasons for restricting accessibility, such as privacy concerns, restrictive company policies, and costliness of the process.

Several efforts to improve transparency and accessibility are already underway. Among these are the Reproducible Evidence: Practices to Enhance and Achieve Transparency (REPEAT) initiative, a program dedicated to the improvement of transparency and reproducibility of database research in the healthcare sector (*Repeat, 2023*). REPEAT contributes to *Wang et al., 2022*, exploring reasons for irreproducibility, responsiveness of corresponding authors, and many more anecdotes in a study attempting to reproduce published results utilizing the same databases as the original investigators. Another example is the Reproducibility Project: Cancer Biology, meant to replicate experiments in cancer biology research papers to analyze their level or replicability (*Davis et al., 2014*). The Reproducibility Project: Cancer Biology demonstrated their work in preclinical cancer biology research replicability by repeating 50 experiments from 23 papers (*Errington et al., 2021b*) and 193 experiments from another 53 papers (*Errington et al., 2021a*), discussing challenges and barriers associated with replicability. In addition, several resources are readily available to aid with the transparency assessment, including but not limited to SciScore (*Menke et al., 2022*), DataSeer (*DataSeer, 2023*), and Ripeta (*Sumner et al., 2021*).

Although some journals and funding agencies have implemented policies to support the public dissemination of research, we know from recent studies in several disciplines that accessible and reproducible research practices are still far from the norm (*Borghi and Van Gulick, 2018*; *Walters et al., 2019*; *Kemper et al., 2020*; *Sherry et al., 2020*; *Smith et al., 2021*). However, to date, no study has examined the degree of accessible and reproducible practices in cardiovascular science publications. Within the domain of cardiovascular science research, this type of analysis is extremely important for holding the latest cutting-edge discoveries in cardiovascular science to a high standard because of the impact on human health. We therefore investigate reproducible reporting practices using an adapted previously published screening process (*National Academies of Sciences, Engineering, and Medicine, 2019*).

With this study, we hoped to identify the prevalence of accessible and reproducible research practices in cardiovascular research. To achieve this aim, we defined what information is necessary to recreate research from a published work (reproducible). We screened randomly selected articles published in 2019 in Circulation, the European Heart Journal, and the Journal of the American College of Cardiology (JACC). We tested, both, that the simple majority, more than half of screened publications, would lack one or more specific criteria that facilitate reproducibility or replicability, and that the vast majority, over 90% of screened publications, would lack one or more specific criteria that facilitate reproducibility or replicability. In addition, we predicted that some study types (e.g. clinical trials) would satisfy significantly more accessibility criteria than other study types. We also predicted that some categories of accessibility criteria would be satisfied significantly more frequently than other categories in both study types and across separate journals (e.g. materials availability vs analysis script availability). We also specified that a lack of specific requirements by journals regarding what information to provide would lead to variability in which accessibility criteria are satisfied.

## Materials and methods
### Sampling plan
To determine the prevalence of accessible and reproducible research practices within cardiovascular literature, data was gathered from a random selection of all published studies in 2019 in the following three leading cardiology journals (*Opthof, 2019*): Circulation, European Heart Journal, and the Journal of the American College of Cardiology (JACC). To limit the scope of our analysis, we only included articles published in the year of 2019. This specific year was selected because it was the most recent full year not influenced by changes in reporting practices due to the COVID-19 pandemic.

The following PubMed search string was used to obtain the full list of articles screened: (('Circulation'[Journal] OR 'J Am Coll Cardiol'[Journal] OR 'Eur Heart J'[Journal]) AND 2019/01/01:2019/12/31[Date – Publication]) NOT (Published Erratum[Publication Type]). This search string retrieved 2786 articles. This list was randomized and the first 639 were screened for our study, and researchers screened the listed articles sequentially. A PRISMA flow diagram (*Appendix 1— figure 1*) of the inclusion and exclusion information is provided in the supplementary materials.

No blinding was involved in this study and each article was screened at least twice by separate individuals. Some researchers screened a higher proportion of articles than other authors, but each author screened at least 180 articles once. Any ambiguities identified during screening were resolved either through additional review, or through discussion among participating researchers to achieve consensus.

We initially screened 400 randomly selected articles, however of those articles only 153 were empirical research articles for which we could calculate accessibility measures. To increase the number of empirical research articles in our study, we therefore conducted a second round of screening. All aspects remained the same during additional screening other than generating the following new PubMed search string: ('Circulation'[Journal] OR 'J Am Coll Cardiol'[Journal] OR 'Eur Heart J'[Journal]) AND 2019/01/01:2019/12/31 [Date – Publication] NOT ((review[Publication Type]) OR (systematic review[Publication Type]) OR (editorial[Publication Type]) OR (comment[Publication Type]) OR ('case reports'[Publication Type]) OR ('Introductory Journal Article'[Publication Type]) OR ('Historical Article'[Publication Type]) OR ('Practice Guideline'[Publication Type])). This search string retrieved 1461 primarily empirical articles. This list was also randomized and the first 239 articles that had not already been screened were added to the first set of 400 screened articles. Researchers screened the new

**Table 1.** Criteria determining accessibility score and criteria-satisfying responses.

| Criteria screened | Criteria-satisfying responses |
|---|---|
| How clearly stated was the study type? | Stated (e.g. editorial comment, clinical trial) |
| Does the article state whether or not materials are available? | Yes the statement says that the materials (or some of the materials are available).* |
| Can you access, download, and open the materials files? | Yes.* |
| Does the article state whether or not data are available? | Yes – the statement says that the data (or some of the data) are available. |
| Can you access, download, and open the data files? | Yes. |
| Are the data files clearly documented? | Yes. |
| Do the data files appear to contain all of the raw data necessary to reproduce the reported findings? | Yes. |
| Does the article state whether or not analysis scripts are available? | Yes – the statement says that the analysis scripts (or some of the analysis scripts) are available. |
| Can you access, download, and open the analysis files? | Yes. |
| Does the article state whether or not the study (or some aspect of the study) was pre-registered? | Yes – the statement says that there was a pre-registration. |
| Can you access and open the pre-registration? | Yes. |
| What aspects of the study appear to be pre-registered? (select all that apply) | Hypotheses, Methods, AND Analysis Plan all available. |
| Does the article link to an accessible protocol? | Yes. |
| What aspects of the study appear to be included in the protocol? (select all that apply) | Hypotheses, Methods, AND Analysis Plan all available. |
| Does the article include a statement indicating whether there were any conflicts of interest? | Any of the following can be selected: Yes – the statement says that there are one or more conflicts of interest. Yes – the statement says that there is no conflict of interest. |
| Does the article include a statement indicating whether there were funding sources? | Any of the following can be selected: Yes – the statement says that there was funding from a private organization. Yes – the statement says that there was funding from a public organization. Yes – the statement says that there was funding from both public and private organizations. Yes – the statement says that no funding was provided. |
| Is the article open access? | Any of the following can be selected: Yes – found via Open Access Button. Yes – found via other means. |

*This criterion was not included for publications for which it was not possible to share materials.

articles sequentially and any remaining non-empirical articles were removed from the list and replaced with an empirical research article.

## Variables

Following a modified version of a previously established coding protocol for reproducible and accessible research practices (*Iqbal et al., 2016*), the following criteria were screened during the evaluation process: type of reported study, pre-registration status, protocol availability, materials availability, data availability, analysis script availability, conflict of interest (COI) status, and open access of the article. An article's 'accessibility score' was calculated as the fraction of several accessibility criteria results with the specific criteria satisfied out of the total possible to be satisfied for that specific study type. See *Table 1* for a list of screening criteria that contributed to the calculation of the accessibility score. Note that for papers without the ability to share Materials (e.g. Meta-analyses), the materials criterion was omitted from the accessibility score calculation. Our article coding form is derived from

*Iqbal et al., 2016* and can be found as Qualtrics.qsf and Word files in the pre-registration (*Campos-Melendez et al., 2021*).

The four criteria defining repeatability – methods (protocol or pre-registration), materials, data, and analysis script availability – are each considered an accessibility category. Each individual criterion in these four categories can be either satisfied or not. For a study to be considered fully 'replicable' all three of the following categories must be available: methods, data, and analysis scripts. For a study to be considered fully 'reproducible', all three of the following categories must be available: methods, data, and analysis scripts. If an article is either partially replicable or partially reproducible it is considered 'partially repeatable'. Definitions of replicability and reproducibility were set based on those introduced by the National Academies of Sciences, Engineering and Medicine (NASEM) (2019). We defined the ability to replicate a study as the attempt to obtain the same results as a study by collecting new data. Reproducing a study describes the attempt to obtain the same results as a study by re-analyzing its data.

## Analysis

Only articles that were fully screened and for which all ambiguities had been resolved were included in the final analysis. Papers written in a language other than English were also excluded, as well as any articles for which we could not access the full text. Articles without empirical data (e.g. review articles, commentaries, and editorials) were evaluated for author location, language, COI, funding statements, and public access of the article itself, but were otherwise not screened for accessibility criteria because these study types by their nature cannot share, for example, materials, data, or analysis code. We also screened whether studies could share materials or not. For example, meta-analyses are empirical research studies, but do not typically have any shareable materials. One such example is a meta-analysis (*Russo et al., 2019*), which was screened as an empirical paper but could not have materials due to the nature of the research conducted. Our analysis does assume, however, that all empirical research articles involve a protocol, the collection or re-use of data, and analysis of that data, and therefore methods, data, and analysis code or scripts should always be available.

To determine how frequently published cardiovascular research reports provide sufficient resources to replicate or reproduce their studies, we quantified the fraction of papers that are either partially replicable or partially reproducible – defined as partially repeatable. For all accessibility criteria, we calculated the proportion of papers in our dataset that exhibit each possible outcome. For example, for the accessibility criterion 'How does the statement indicate the materials are available', we calculated the proportion of papers that state that materials are available through an online third party repository, a personal or institutional webpage, supplemental information hosted by a journal, or upon request from authors. We also calculated the distribution of accessibility scores across our full dataset and further collected data regarding the location of the corresponding author. We use standard deviations to show the variability of the data.

To test for inconsistent, or variable, levels of accessibility, we examined our data using multi-way ANOVAs. Our independent variables included 'study type' and 'journal' and our dependent variable was the accessibility score. For the hypotheses addressing differences between the types of studies, journals, and accessibility scores using multi-way ANOVA, p-values <0.05 were considered to be statistically significant.

To evaluate the relationship between accessibility criteria (Materials, Methods, Data, and Analysis Code) and study types, we performed several sequential chi-squared tests of proportions. In cases with 0 samples in a group, we instead performed Fisher's exact tests. Tests were performed using R. We compared the proportion of papers that did and did not allow access to those four categories of information and resources, across all four categories. We also used chi-squared tests to further compare the proportion of papers that provided access to each category of information and which study types shared more or less of a specific type of information compared to other study types. We considered p-values ≤0.05 to indicate significant differences in proportions. We used Bonferroni correction to adjust the p-value for the number of tests performed. Note that this analysis differs slightly from that proposed in our pre-registration: Chi square Automatic Interaction Detection (CHAID).

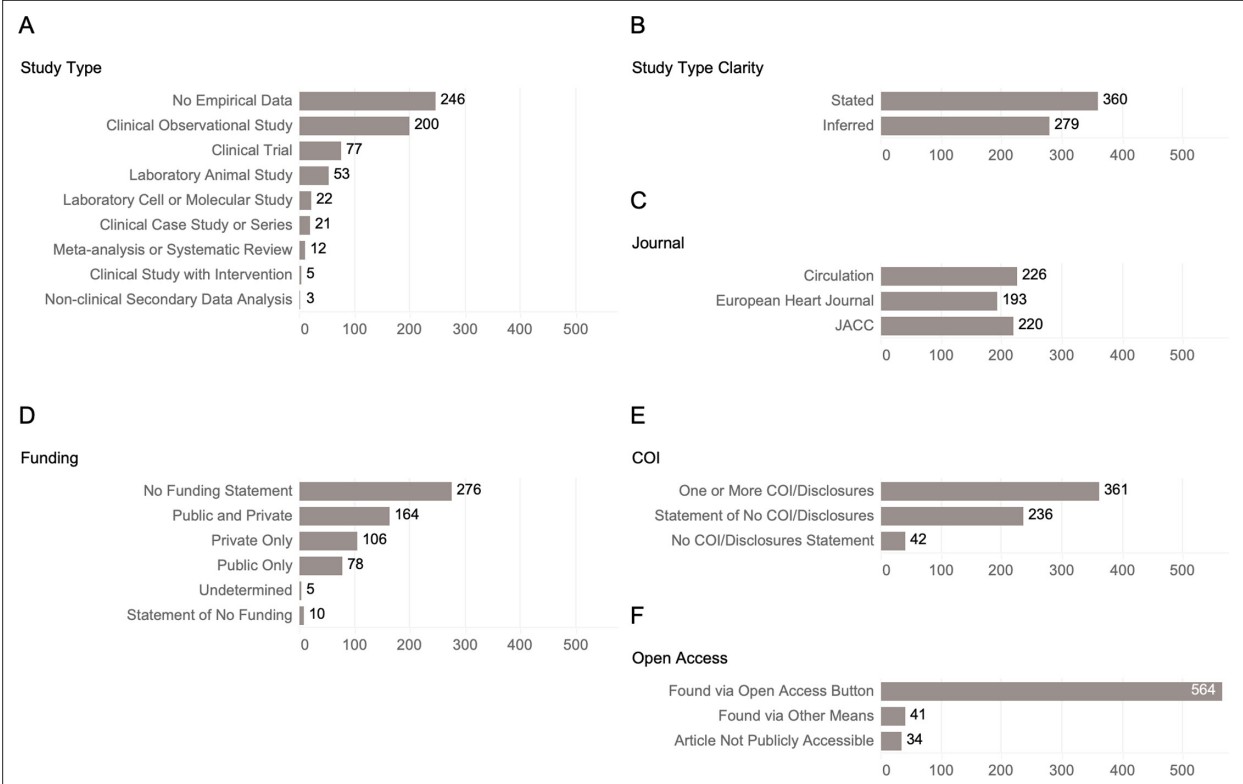

**Figure 1.** Distribution summary of all screened publications. Summary of all screened publications on their (**A**) study type, (**B**) study type clarity, (**C**) journal in which it was published, (**D**) presence of funding source statement, (**E**) presence of conflict of interest (COI) statement, and (**F**) open access status.

## Pre-registration

This study was pre-registered through the Open Science Foundation and can be accessed at https://doi.org/10.17605/OSF.IO/QFSTH (*Campos-Melendez et al., 2021*).

After obtaining our data, not all exploratory analyses documented in the pre-registration were performed. We did not test the relationship between the articles described as accessible and those actually accessible due to the limited number of articles that exhibited large enough accessibility scores, nor did we perform textual analysis on the COI, funding statements, or the way resources were being shared. We also deviated from evaluating the relationship between corresponding author location and accessibility score given the low sample size for most countries. We also did not perform additional analyses with the accessibility criteria organized as a hierarchy, because of the subjectivity in determining a ranking of screening criteria. We also deviated from the pre-registration by using sequential chi-squared tests instead of the CHAID. CHAID analysis builds a predictive model using a classification tree with multiple levels. In our case, we only compare data across two levels and there is no hierarchical basis for a decision tree. CHAID was therefore an inappropriate method for our study. Lastly, we extended our data collection beyond what we initially proposed in our pre-registration so that we could have more empirical research articles in our dataset.

## Results

We screened a similar proportion of 2019 articles from each of the three top cardiovascular journals: Circulation, European Heart Journal, and the Journal of the American College of Cardiology (226, 193, and 220, respectively, *Figure 1C*). Over half (247 out of 400) of the randomly selected articles in our initial screening were review articles or other article types that contained no empirical data (*Figure 1*). We ultimately screened a total of 639 articles, of which 393 were empirical research studies. Approximately half of the papers (56%) had easily identifiable study types while the remaining papers required more substantial reading to determine their study type (*Figure 1B*). 43% (276 out

of 639) of the articles did not have a funding statement (*Figure 1D*). 93% of articles included a COI statement, and 57% stated they had a COI (*Figure 1E*). Notably, 7% of articles did not have any COI statement at all. Additionally, over 90% of articles were publicly accessible (605 out of 639, *Figure 1F*).

## Cardiovascular research publications rarely make their resources available

Looking only at articles that included empirical research (393 out of 639 total articles), the majority had had no pre-registration statement (286) or no linked and accessible protocol (383; *Figure 2A*). Of the papers that had a pre-registration statement, nearly all of them had accessible and openable pre-registrations (106 out of 107; *Figure 2A*). Among articles that had an openable and accessible pre-registration or protocol, almost all articles included methods (106 out of 106, 9 out of 10), approximately half included hypotheses (44 out of 106, 7 out of 10), and less than half included analysis plans (27 out of 106, 5 out of 10; *Figure 2A*). We define a publication to have all pre-registration aspects if hypotheses, methods, and analysis are pre-registered. Of the 393 screened publications that have empirical research, 53 had 1 out of 3 of the aspects, 35 had 2 out of 4 of the aspects, and 18 studies had all three aspects.

Only study types considered empirical research in which materials are theoretically available are summarized. Only 14% of articles made their materials available (56 out of 393), 31.8% of articles made their data available (125 out of 393), and 10.9% made their analysis scripts available (43 out of 393; *Figure 2B*). Across the articles that stated materials, data, and analysis scripts were available, it was most common for them only to be made available upon request from the authors (*Figure 2B*). Across the articles that stated materials, data, and analysis scripts were available, very few studies made the materials and data readily available for downloading or opening (1, 13) and only one study's analysis scripts were readily downloadable (1 out of 43; *Figure 2B*). However, the majority of data that was downloadable or openable was also clearly documented (9 out of 13; *Figure 2B*).

The majority of cardiovascular publications are not readily replicable or reproducible. We predicted that the majority of publications will not provide sufficient resources to replicate or reproduce their studies. We tested both (1) that the simple majority, more than half of screened publications, will be lacking one or more specific replicability or reproducibility criteria, and (2) that the vast majority, over 90% of screened publications, will be lacking one or more specific reproducibility or replicability criteria. We found that 49.6% (195 out of 393) of empirical research studies were not partially reproducible and 49.4% (194 out of 393) were not partially replicable. 2% (7 out of 393) empirical research papers were fully reproducible, but only 5 were fully replicable (*Figure 3*). Therefore, the simple majority of empirical research articles were partially reproducible or replicable, but the vast majority of empirical research studies were neither fully reproducible nor fully replicable.

## Accessibility varies across journals and study types

With the exception of one paper, the entire dataset exhibited low accessibility with scores lower than 0.6; the majority of articles (144 out of 393) had accessibility scores within the 0.20–0.29 range and 18 articles had scores below 0.1. Only one article exhibited an accessibility score of 0.6 or greater (*Figure 4A*).

We also predicted that the level of accessibility will be inconsistent across several dimensions. Specifically, we predicted that average accessibility score would vary across study types and potentially also across journals, depending on the publisher's requirements. For example, if clinical trials must be registered and open access, clinical trial publications will satisfy more of the article coding form criteria than other studies. We tested this hypothesis by calculating an accessibility score for every article: the sum of the satisfied screening criteria divided by the total possible satisfiable criteria. We then calculated the average and standard deviation in accessibility scores across study type (*Figure 4C*). Overall accessibility scores were low across all study types, and standard deviations were large, indicating high variability across individual publications. We found that clinical trials had the highest average accessibility score (0.39), and clinical case studies or series had the lowest accessibility scores (0.08). Average accessibility scores were significantly different across study types (ANOVA, d.f. 7, $F=36.37$, $p=5.500E-5$).

We also predicted a significant difference in the number of accessibility criteria satisfied across journals due to differences in reporting policies. The largest proportion of articles screened were obtained

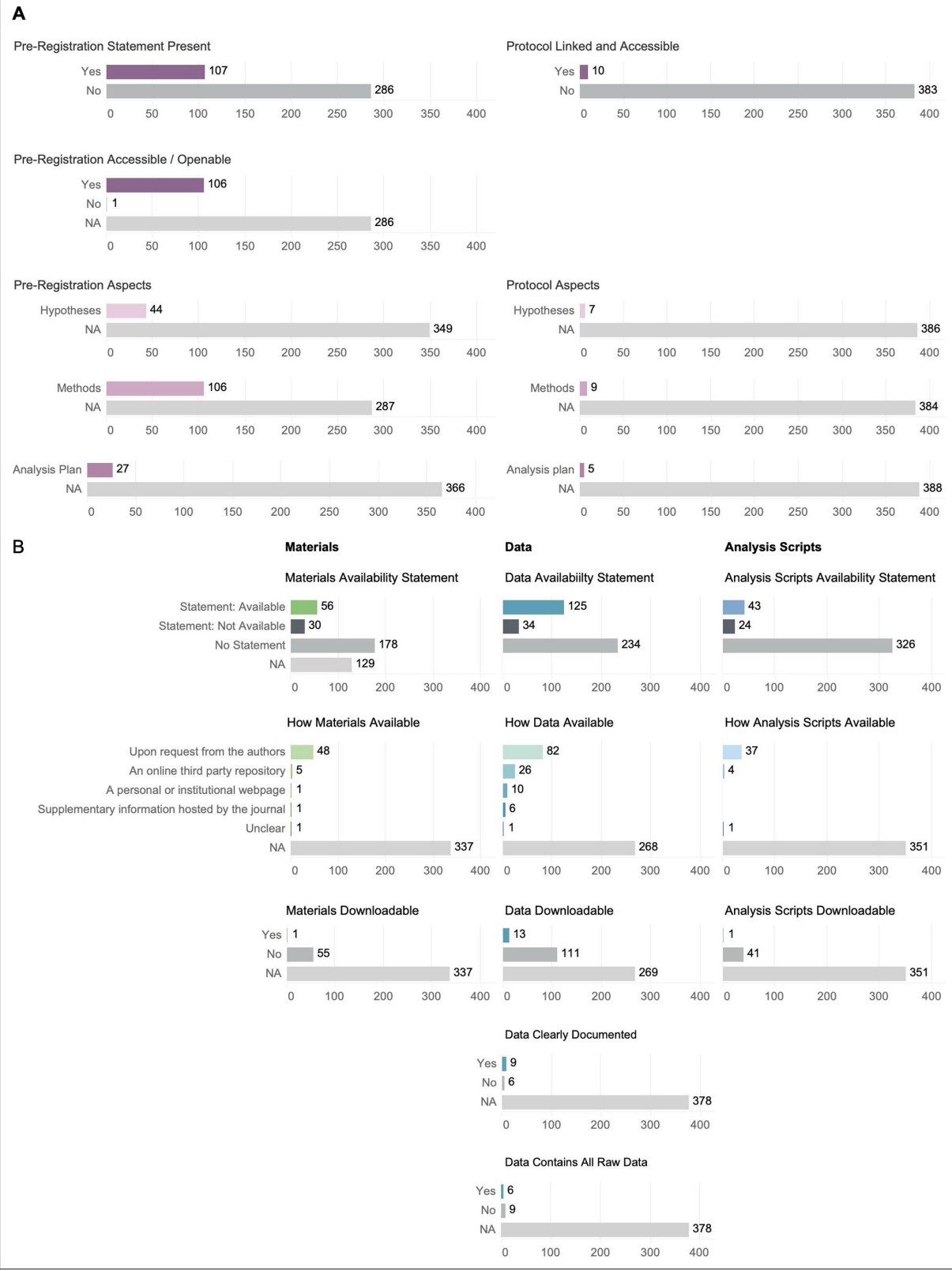

**Figure 2.** Summary of screened papers for pre-registration, protocol, material, data, and analysis script availability. Only study types considered empirical research in which materials are theoretically available were summarized. (**A**) Summary of the presence and accessibility of pre-registrations and protocols and the summary of components (hypotheses, methods, and analysis plan) of pre-registration and protocol for papers that had a pre-registration or protocol statement. N/A represents papers that could not answer the criteria because data, materials, and analysis plans were not

*Figure 2 continued on next page*

*Figure 2 continued*
available to begin with (top panel). (**B**) Summary of papers and material, data, and analysis script availability. For papers that had a statement, how materials, data, and analysis script were available, whether they were accessible, and whether it was clearly documented or present in its entirety are also summarized.

from Circulation with a total number of 226 articles, and 74% of those were empirical research papers (168 out of 226). 63% of JACC articles (138 out of 220) and 45% of Circulation articles (87 out of 193) were empirical research for which we could calculate an accessibility score. After computing the average accessibility score for all three journals, Circulation had the highest average accessibility score (0.34), followed by the Journals of the American College of Cardiology (0.24), then the European Heart Journal (0.19) (*Figure 4B*). Accessibility scores vary significantly across journals (ANOVA, d.f. 2, $F$ = 80.07, p = 1.230E−2). There was not a significant interaction between study type and journal (p = 0.284).

## Categories of accessible resources varies across study types

We also predicted some categories of resources will be accessible significantly more frequently than other categories (e.g. materials availability vs analysis script availability). We found that there was a significant difference in the proportion of papers that shared specific categories of resources ($X^2$ = 55.2, p = 6.17e−12). We then further identified whether different study types exhibited differences in the proportions of shared resources, for example, whether there was a difference in the proportion of papers that have accessible data depending on study type. Note that we did not include Clinical Studies with Interventions, Meta-analysis or Systematic Reviews, Non-clinical Secondary Data Analysis, or Laboratory Cell or Molecular Studies in this analysis due to their small sample size. For all four

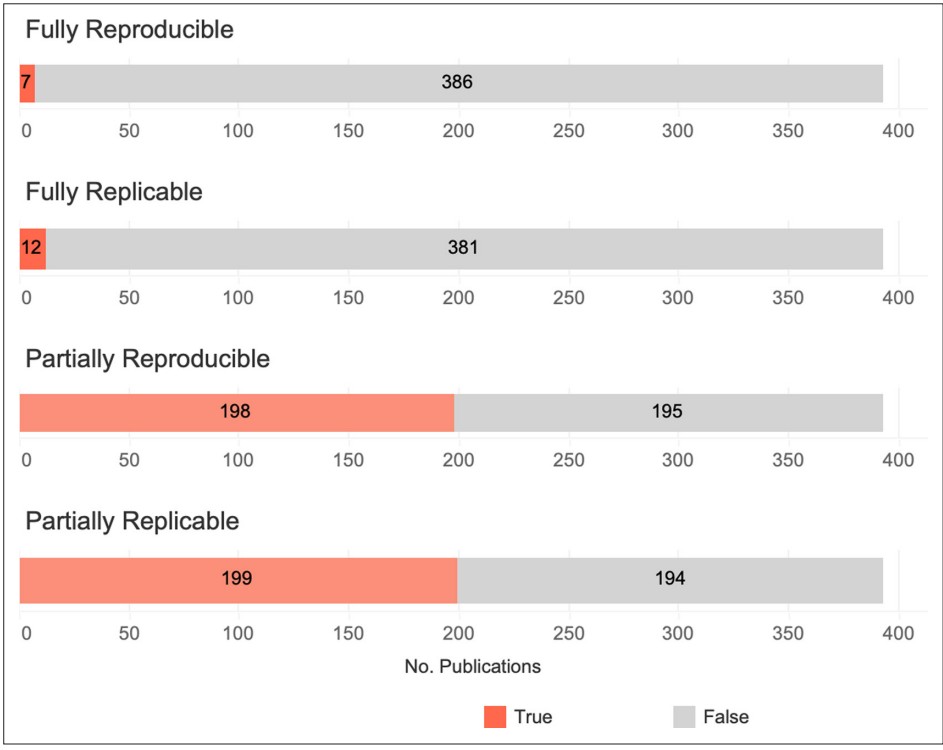

**Figure 3.** Summary of potential replicability and reproducibility of empirical research studies (*n* = 393 studies). An article is considered 'partially replicable' if any of material availability, analysis script availability, and methods criteria are satisfied and 'fully replicable' if all three criteria are satisfied. An article is considered 'partially reproducible' if any of data availability, analysis script availability, and methods are satisfied and 'fully reproducible' if all three criteria are satisfied. Note that these data describe the *potential* for a study to be partially or fully reproduced or replicated based on the availability of the study's resources (methods, materials, data, and code), not whether the study was itself replicated or reproduced.

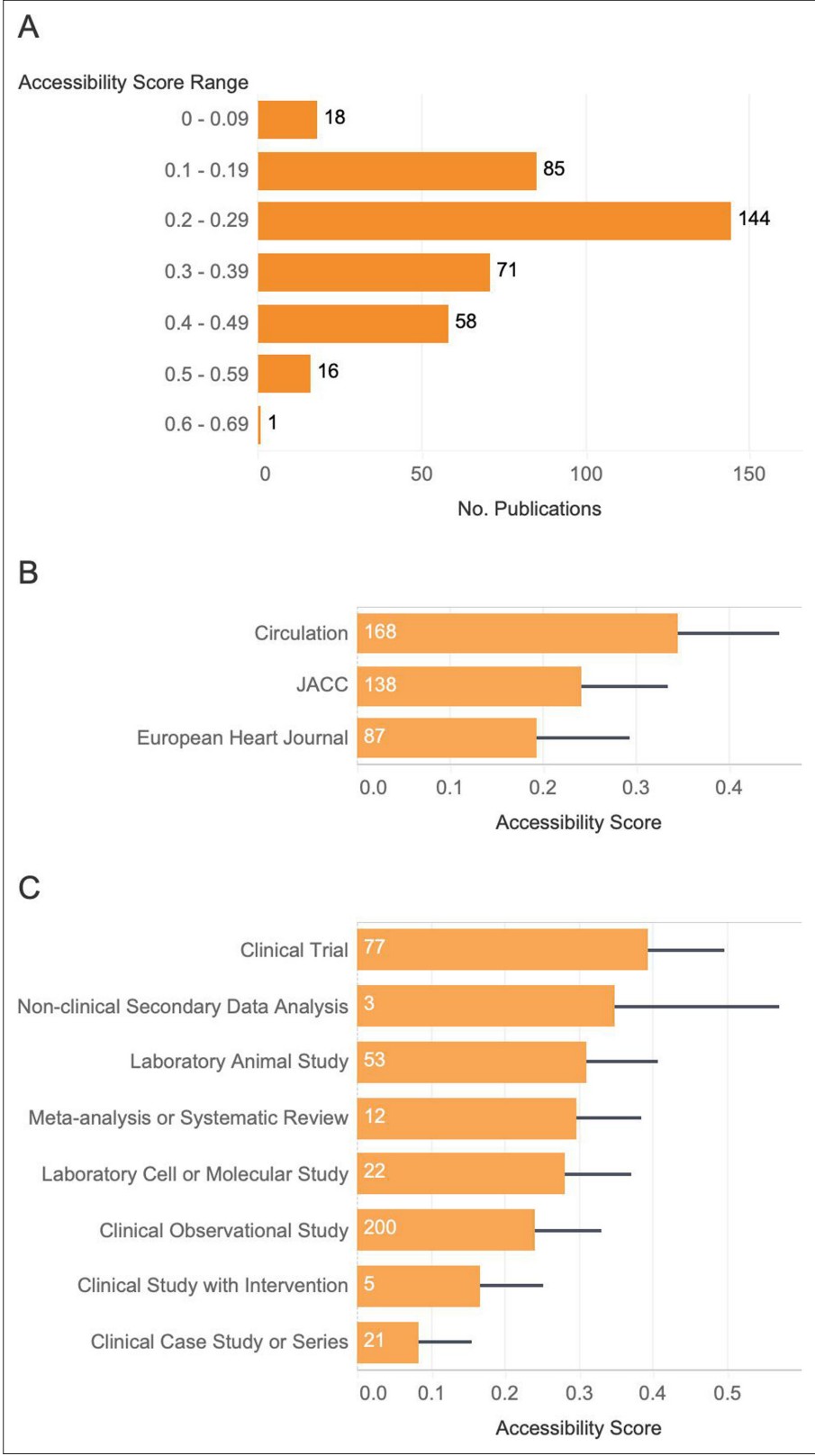

**Figure 4.** Accessibility score distribution accross the dataset, by study type and by journal type. (**A**) Accessibility score distribution across the entire dataset. The number of articles represented by each bar falls within the specified accessibility score range. No articles obtained an accessibility score fraction greater than 0.60. (**B**) Average accessibility scores across all articles screened, based on study type. Error bars correspond to standard

*Figure 4 continued on next page*

*Figure 4 continued*

deviation of the collected data points. Sample sizes are noted at the base of each bar. All scores are based on a total possible score of 1. (**C**) Average accessibility score by journal type. Screened articles were obtained from three different journals, including Journals of the American College of Cardiology (JACC), European Heart Journal from the European Society of Cardiology, and American Heart Association (AHA) Circulation. The standard deviation bars are indicative of the range of distribution of the obtained accessibility score fractions for each journal. Sample sizes are noted at the base of each bar. All score fractions are based out of a total possible score of 1.

shareable resources (Materials, Methods, Data, and Analysis), we found a significant difference across study types (*Table 2*).

To determine specifically how study types differ in which resources they share, we performed multiple chi-squared tests with Bonferroni correction, comparing each study type with every other study type for a given resource (*Figure 5*; *Table 3*). We found that a significantly higher proportion of laboratory animal studies stated that they would share materials than clinical observational studies or clinical trials. We also, as expected, found a significantly higher proportion of clinical trials shared their methods compared to any other study type. With regard to data sharing, we found that a significantly higher proportion of laboratory animal studies shared their data than any other study type. Lastly, a significantly higher proportion of laboratory animal studies shared their analysis code compared to clinical observational studies and clinical trials.

## Discussion

We reviewed 639 papers that were published in 2019 in three of the top cardiovascular research journals to determine how accessible, and therefore reproducible, their research was. Our intention with this study was not to diminish existing studies, but to share current practices with the cardiovascular field and identify ways to improve those practices to enhance reproducibility and transparency in the future. We hope to identify opportunities for journals and scientists to adapt their practices to further reproducible science and encourage the use of higher standards and consistent formats, among cardiovascular scientific literature.

In general, we found that the simple majority, but not the vast majority, of publications are lacking one or more of the resources (materials, methods, data, or analysis scripts) to replicate or reproduce a study. Only 5 out of 393 provided sufficient resources to fully replicate their work, and only 7 out of 393 provided sufficient resources to reproduce their work. Although there were statistically significant differences in accessibility scores across study types and journals, overall, accessibility scores were consistently low.

In an effort to increase replicability and reproducibility of future published cardiovascular literature, there are some initiatives that could be taken. Journals have implemented practices to incentivize researchers to share their materials, methods, data, and analysis scripts, such as open science badges (*Kidwell et al., 2016*), reporting checklists (*Han et al., 2017*), and checking articles for reproducibility (*Organic Syntheses, 2023*) or replicability (*Nosek et al., 2015*). Our findings suggest increasing these efforts and expanding them to funding agencies would help promote open science practices across the field.

**Table 2.** p-values of chi-square tests of study type for each category of resource.
We tested for a significant relationship between study type and availability of resources. 'Yes' refers to a paper including the resource and 'No' refers to a paper not including the resource. Observed values were compared to expected values for each category of resource.

|  | Yes | No | Study type chi-square p-value with Bonferroni correction |
|---|---|---|---|
| Materials | 46 | 308 | 3.05E−13 |
| Methods | 107 | 247 | 4.91E−35 (case studies omitted) |
| Data | 107 | 247 | 2.31E−12 |
| Analysis | 36 | 318 | 1.58E−12 |

**Figure 5.** Proportion of papers with presence (yes) or absence (no) of specific accessibility criteria (Material, Methods, Data, and Analysis Code) for specific study types. Methods presence is determined by presence of a pre-registration or linked protocol. Presence is indicated in green, absence in yellow. For each value, both the percentage and the count for that category and study type are shown.

## Publication accessibility

In collecting data on the fraction of papers that were publicly accessible, we found that only 34 out of 639 articles (5%) were not publicly accessible (*Figure 1F*). Although this is a fairly low fraction of inaccessible articles, a study's publication is in many ways the most tangible output of the research, and having publicly accessible publication should be a minimum standard for research. The NIH's public access policy has been instrumental in ensuring public access to published research reports, but reports funded purely through private sources are not currently under the same reporting mandate.

## Study type

It is notable that over half of the 400 papers we initially randomly selected for screening were non-empirical research, e.g. reviews, editorials, and commentaries. Although these emissions may be effective ways to communicate quickly and effectively with larger audiences, they may not go through the same rigor of review process. We also occasionally screened articles that were simplified summaries of the original research study, geared toward a lay audience. This type of article is valuable in that it makes research more accessible to a broader audience; however, because methods were absent and results were condensed, it was ambiguous whether the studies being reported in these articles had undergone full scientific review. Readers run the risk of assuming that these articles describe a full scientific story as opposed to a news highlight.

In addition, publications of replication studies, or studies that included a replication study, were virtually absent from our dataset (4 out of 393 empirical research publications) – suggesting the field of cardiovascular research puts very little emphasis on replication work. Of concern, several studies also stated outright that their data was not available for replication, stating for example 'The investigators will not make the data, methods used in the analysis, and materials used to conduct the research available to any researcher for purposes of reproducing the results or replicating the procedure.' Although there can be legitimate reasons why data sharing is not possible, these statements are typically not accompanied by any justification.

**Table 3.** p-values for chi-squared tests and Fisher's exact tests comparing the presence or absence of specific accessibility categories (Materials, Methods, Data, and Analysis Code) for each study type compared to every other study type (Clinical Case Study or Series, Clinical Observational Study, Clinical Trial, and Laboratory Animal Study).

p-values significant at an alpha level of 0.05 with Bonferroni correction are shown in bold. Use of Bonferroni correction to adjust the p-values for multiple comparisons resulted in some p-values being greater than 1. Given that a probability greater than 100% cannot occur, any values greater than 1 were adjusted to a ceiling value of 1.

| | Materials | Methods | Data | Analysis Code |
|---|---|---|---|---|
| Clinical Case Study or Series vs Clinical Observational Study | 1.0 | 1.0 | 1.0 | 1.0 |
| Clinical Case Study or Series vs Clinical Trial | 1.0 | **3.14E−13** | 0.34 | 1.0 |
| Clinical Case Study or Series vs Laboratory Animal Study | **0.04** | 1.0 | **1.86E−5** | 0.18 |
| Clinical Observational Study vs Clinical Trial | 1.0 | **1.35E−26** | 0.25 | 1.0 |
| Clinical Observational Study vs Laboratory Animal Study | **1.58E−12** | 1.0 | **1.74E−11** | **2.66E−12** |
| Clinical Trial vs Laboratory Animal Study | **1.42E−3** | **3.96E−19** | **2.07E−3** | **1.92E−3** |

## Materials, methods, data, and analysis script sharing

Although data was shared more frequently than materials or analysis scripts, there were still only a total of 13 out of 393 empirical research papers for which data was readily openable to a reader. We acknowledge that authors will often have legitimate reasons for not being able to share resources, including patient privacy. However, it has also been shown that patients are in general very willing to make their data available to further research (*Seltzer et al., 2019*; *Kim et al., 2019*). We advocate for research studies actively seeking consent from human subject participants to make their data available and if that consent cannot be obtained to specify that justification for not sharing data in their report.

Although basic research studies more frequently shared materials, data, and analysis scripts than clinical trials, that sharing was frequently 'upon request from the authors', which previous studies have shown to be hit-or-miss in terms of yield (*Hardwicke et al., 2018*). Clinical trials had by far the most consistent availability of a resource category: methods, in the form of pre-registrations. Because pre-registrations are mandated by the FDA to conduct a clinical trial, our results suggest that resource-sharing requirements by funding or approval agencies are effective means of changing practices. It should be noted however, that a recent study by *Goldacre et al., 2019* showed that in trials with pre-registrations, only 76% of pre-specified trial outcomes were correctly reported. Therefore, even when trials are pre-registered variable switching is common when reporting results.

## COI and funding statements

Publications are a critical form of communicating research, and the content and results of publications are often prioritized to the point that COI and funding statements are easy to overlook. As a community, we are often more interested in the results described in publications than the process behind getting them. Personal interests and finances are undeniably a part of experimental integrity and can impact experimental design and therefore COI and funding statements should be given as much attention as other components of the publication. Although it was beyond the scope of this study to perform a rigorous analysis of COI and funding statements, our screening process did reveal numerous cases of ambiguous COI and funding declarations. In *Table 4*, we capture both problematic and positive examples of COI and funding statements. For example, many COI and funding statements were vague in that they lacked details on how different funders of interests specifically influenced the study, which is important for interpretation of the results. As another example, we also identified 'Disclosures: None' as a problematic statement, because it could be interpreted as the authors having no disclosures or that the authors declined to list their disclosures. The relationship between funders and COIs is also ambiguous. Authors frequently listed funders and declared 'no conflict of interest'; however, funders frequently do have an interest in and impact on the study and therefore represent a COI. If nothing else, it is in the interest of researchers to produce compelling results to maintain good relations and receive future funding from agencies, even if they were funded solely through public organizations.

To avoid these issues, we advocate that journals require more complete and standardized COI and funding information that recognizes their overlap, including: a clear description on how each funding entity supported the work including whether they influenced the experiments, analysis, or dissemination of the work. We recommend that journals require authors to complete a consistent and transparent funding and COI questionnaire or decision tree that is clearly and prominently associated with the article.

## Authorship

Although we did not have sufficient articles from different countries of corresponding authors to determine whether there was a significant difference in accessibility practices across countries, we did see a wide range in accessibility score values. In terms of reporting, it was in general very straightforward to identify the corresponding author of an article. However, sometimes corresponding authorship was ambiguous in that no contact information was given, or was listed as, 'Published on behalf of […]. All rights reserved. The Author(s) 2019. For permissions, please email: […].' In these circumstances, it is unclear who the corresponding author is and who is ultimately responsible for questions regarding the research.

**Table 4.** Summary table of problematic and informative COI and funding statement examples that we repeatedly came across. Ambiguous or problematic statements are in red and clear statements are in green.

**Conflict of interest (COI) statements**

| Example | Comment |
|---|---|
| When the COI is mixed with funding when disclosures are listed as 'none'. | While funding can be considered a conflict of interest, the statements should be carefully separated if 'disclosures' and 'funding' are listed separately. Support for a specific author may not always be directly funding the project but may influence the project as a conflict of interest. |
| 'Conflict of interest: the author has received fees or grants from COMPANY.' | 'Fees' and 'grants' are two different elements and should be clarified as both can influence a study. It is unclear whether these fees or grants funded the published study and how they influenced the study if at all. |
| 'Both authors have reported that they have no relationships relevant to the contents of this paper to disclose.' 'The authors have reported that they have no relationships relevant to the contents of this paper to disclose.' 'Conflicts of interest: none.' 'Disclosures: none.' | This statement suggests that there are potential conflicts of interests and leaves the reader wondering if there are possible conflicts of interests left out. It does not leave room to be confident that there are no conflicts of interests. |
| 'Dr. NAME has received research grants from NAME; and has received honoraria for NAME. Dr. NAME has received research grants from NAME. Drs. NAMES are founders of COMPANY and as such have received modest honoraria from COMPANY.''Dr. NAME is related through family to a member of COMPANY but neither she, nor her spouse, nor children have financial involvement or equity interest in and have received no financial assistance, support, or grants from the aforementioned.' | It is highly ambiguous as to whether this is a funding or COI statement. It is unclear if funders played any role in designing or performing the experiment. |
| There is no COI but there is an acknowledgement. | Acknowledgements and COI should be separate sections as these two sections have different purposes. Including COI as acknowledgement allows COIs to be easily overlooked. |
| No COI or funding statement. | COI and funding statements provide additional factors that can impact a study outside of experimental factors. This information should always be included to fully inform readers, particularly when concerns are raised about certain studies or when the information is applied in real-world applications. |
| 'NAME served as the Guest Associate Editor for this paper.' | Although many authors will not serve as editors for the journals they are applying to, it is helpful to acknowledge when that is the case and the potential conflict of interest. |
| The authors specifically state that they have no conflicts of interest. | This is a clear and definitive statement that there are no conflicts of interests and readers are not left wondering if there are conflicts of interests that are not mentioned. |
| Itemization of funders with explicit listing of the ways funders did not contribute to the study. 'The authors have reported that they have no relationships relevant to the contents of this paper to disclose.' | We believe this is an excellent way of recognizing that funding can be a form of conflict of interest. The authors also specifically state that they have no conflict of interests. |

**Funding statements**

| Example | Comment |
|---|---|
| Funding in acknowledgements. | Funding should be separate from acknowledgements. When funding is included in acknowledgements, it is easily overlooked. |
| Long list of affiliations without any statement that the list is funding or COI. | Funding and COI should be considered as an opportunity to share how factors outside of the experiment influenced the study. Listing affiliations is not sufficient and should be followed by explanations of why they are listed. |
| Funding statement system, particularly when papers list 'funding on page PAGENUMBER'. | Listing 'funding on page PAGENUMBER' is ambiguous, leading to issues with finding the funding statement. We experienced cases where we either could not find the funding statement, or it was difficult to access. Funding statements should be listed with their respective articles. |
| Funding statement found on pages outside of their respective articles 'Sources of Funding, see page PAGENUMBER.' | Funding statements should be directly associated with their corresponding article. Although including a funding statement elsewhere in a journal issue is better than no funding statement, the location is distant and disconnected from its corresponding article. If a pay wall is present, access may differ between the funding statement and the original article. |

*Table 4 continued on next page*

*Table 4 continued*

**Funding statements**

| | |
|---|---|
| 'Acknowledgements: Dr. NAME is a recipient of a grant from the ORGANIZATION in support of SPECIFIC research.' | We believe that funding should be separate from acknowledgements, as these two sections have their own purposes. |
| 'Sources of Funding: none.' 'Disclosures: Drs. NAMES received modest consulting fees from COMPANY for the conduct of this research. NAME is funded by grants from COMPANY. Dr. NAME reports a charitable grant from the ORGANIZATION, and personal fees from COMPANY. The other authors report no conflicts.' | There are no funding sources listed, but numerous avenues of fees, grants, and more are then listed as disclosures. It is not clear that these fees and grants did not influence the study in any way, including whether those fees and grants were used to partially fund the study. |
| Under 'sources of funding', no sources of funding are listed, but the COI statement refers to (explicit) statements that describe funding for the study. | There should not be discrepancies between reports on funding and COI. This makes it difficult for the reader to gauge how factors are truly impacting the study. |
| Funding and COI are condensed into a single section below author associations. Funding statement and COI are listed together as 'Footnotes'. | COI, funding, and author associations should be listed separately for easy understanding. By listing COI, funding, and author associations together, it is difficult to understand who impacts the study in what ways. |
| 'Dr. NAME is supported by FUNDING from the COMPANY/ORGANIZATION. The funding source had no role in the design and conduct of the study; collection, management, analysis, and interpretation of the data; preparation, review, or approval of the article; and decision to submit the article for publication.' | This statement clearly acknowledges how funders can play a role in a study and explicitly states that funders had no role in the experiment and publication. The reader is not left wondering if there are additional conflicts. |

## Limitations

Although we attempted to achieve completeness and consistency in screening by ensuring each article was screened by two separate individuals, we acknowledge we may have missed or misinterpreted statements in screened articles. For example, in capturing the type of funding for a study as public, private, or a combination of both, we may have misidentified the funding type for some organizations, especially for foreign funding bodies. Another example would be identifying whether the study type was clearly stated versus inferred with further reading of the article. These determinations are subjective and may vary with individuals' experience with different study types and interpretation of language used in articles. Note that regular readers of publications will face the same challenges.

Furthermore, the accessibility score we used could be further developed to better represent the diversity of publications. For example, there may be types of studies that we identified as not providing materials, because no materials statement was given, but for which all materials may already be adequately described in the text and therefore actually readily available. In these cases, our accessibility score is not accurate. Case studies and case series are the study type most likely to be affected by this limitation due to their limited data and analysis. Only 21 out of 393 empirical research studies (5%) were case studies or case series. Similarly, our current screening criteria only gives credit to papers that state they share a protocol and that protocol is actually linked. Our screening protocol does not capture articles that have a protocol availability statement but do not provide a link.

In the future, the accessibility score could be improved to be more specific to different article types. For example, case studies typically represent patient cases where replicating or reproducing the study would not necessarily be expected. Lastly, this study also only used data from some of the highest ranking cardiovascular research journals. These journals are likely to have higher reporting standards than other journals in the field, so our results are likely to overestimate the reporting and sharing practices of publications in cardiovascular research in general.

## Future work and conclusion

The data collected from 639 screened papers provides numerous future directions for not only exploratory analysis on the existing dataset, but also for new projects assessing accessibility and reproducibility of scientific literature. The accessibility scores we calculated are a rough, quantitative estimate of an article's actual accessibility and further work is required to more fully describe how accessible and reproducible cardiovascular literature is. For example, future work could identify which criteria are the biggest needs for the field, and then evaluate the quality of an article's accessibility by weighting

those criteria or organizing the criteria into a hierarchy of importance. Future studies could also investigate text excerpts describing how resources are or are not being made available to determine the causes that promote or undermine accessible research practices.

Our study shows that there is a high degree of variability in the resources cardiovascular research publications make available – across study types and journals. Universally, however, publications almost never provide sufficient materials, protocol information, data, or analysis scripts for another group to fully replicate or reproduce their work. When federal policies mandate the open research practices, such as ensuring publications are publicly accessible or clinical trials are pre-registered, these practices are adopted. To ensure the highest caliber of research in the future, we urge journals and funding agencies to require higher standards in their materials, protocol, data, and analysis script sharing – regardless of the study type or the funding source.

## Additional information

### Funding

| Funder | Grant reference number | Author |
| --- | --- | --- |
| National Institutes of Health | R25HL147666 | Gabriel O Heckerman Arely Campos-Melendez |
| American Heart Association | 18UFEL33960207 | Eileen Tzng |

The funders had no role in study design, data collection, and interpretation, or the decision to submit the work for publication.

### Author contributions
Gabriel O Heckerman, Eileen Tzng, Arely Campos-Melendez, Data curation, Software, Formal analysis, Validation, Investigation, Visualization, Methodology, Writing – original draft, Writing – review and editing; Chisomaga Ekwueme, Software, Validation, Investigation, Visualization, Methodology; Adrienne Mueller, Conceptualization, Resources, Supervision, Funding acquisition, Investigation, Methodology, Writing – original draft, Project administration, Writing – review and editing

### Author ORCIDs
Gabriel O Heckerman ⓘ https://orcid.org/0000-0001-6420-8909
Eileen Tzng ⓘ https://orcid.org/0000-0002-7513-9192
Arely Campos-Melendez ⓘ https://orcid.org/0000-0002-0322-1170
Adrienne Mueller ⓘ https://orcid.org/0000-0001-9161-5323

### Decision letter and Author response
Decision letter https://doi.org/10.7554/eLife.81051.sa1
Author response https://doi.org/10.7554/eLife.81051.sa2

## Additional files

### Supplementary files
MDAR checklist

### Data availability
All materials, data, and analysis scripts associated with this study are available on Open Science Framework.

The following dataset was generated:

| Author(s) | Year | Dataset title | Dataset URL | Database and Identifier |
|---|---|---|---|---|
| Tzng E, Heckerman G, Campos-Melendez A, Mueller A, Ekwueme C | 2025 | Transparency of research practices in cardiovascular literature | https://doi.org/10.17605/OSF.IO/FUDKA | Open Science Framework, 10.17605/OSF.IO/FUDKA |

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

# Appendix 1

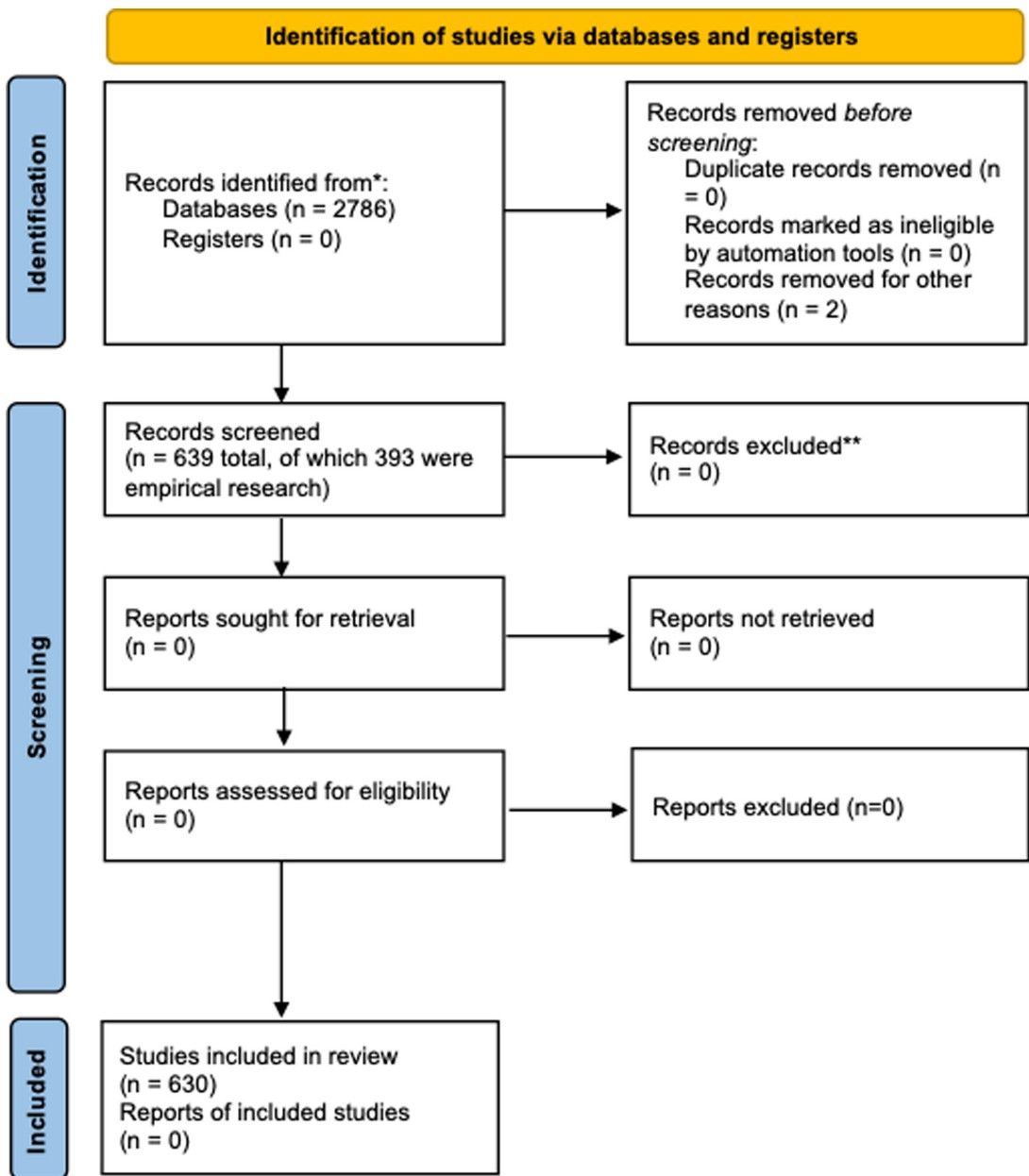

**Appendix 1—figure 1.** PRISMA 2020 flow diagram showing the inclusion and exclusion criteria for our screening process (*Page et al., 2021*). *Consider, if feasible to do so, reporting the number of records identified from each database or register searched (rather than the total number across all databases/registers). **If automation tools were used, indicate how many records were excluded by a human and how many were excluded by automation tools.

