## [Editor Report]

This paper in the field of metascience reports important findings on the levels of accessibility and reproducible research practices in the field of cardiovascular science. As such, it provides a solid benchmarks against which future work could be assessed. The article is of broad interest to basic and clinical cardiovascular scientists.

---

## [Decision Letter]

**Decision letter after peer review:**

Thank you for submitting your article "Accessibility and Reproducible Research Practices in Cardiovascular Literature" for consideration by *eLife*. Your article has been reviewed by 2 peer reviewers, and the evaluation has been overseen by a Reviewing Editor and Matthias Barton as the Senior Editor. The following individuals involved in review of your submission have agreed to reveal their identity: Timothy M Errington (Reviewer #1); Dorothy VM Bishop (Reviewer #2).

Essential Revisions (for the authors):

The analysis states that to be fully reproducible, publications must include sufficient resources (materials, methods, data and analysis scripts). But how about cases where materials are not required to reproduce the work? In line 128-129 it is noted that the materials criterion was omitted for meta-analyses, but what about other types of study where materials may be either described adequately in the text, readily available (eg published questionnaires), or impossible to share (e.g. experimental animals). To see how valid these concerns might be, the first 4 papers were assessed in the deposited 'EmpricalResearchOnly.csv' file. Two had been coded as 'No Materials availability statement' and for two the value was blank.

Study 1 used registry data and was coded as missing a Materials statement. The only materials that might be useful to have might be 'standardized case report forms' that were referred to. But the authors did note that the Registry methods were fully documented elsewhere.

Study 2 was a short surgical case report – for this one the Materials field was left blank by the coder.

Study 3 was a meta-analysis; the Materials field was blank by the coder

Study 4 was again coded as lacking a Material statement. It presented a model predicting outcome for cardiac arrhythmias. The definitions of the predictor variables were provided in supplementary materials. It is not clear what other materials might be needed.

These four cases suggest that it is rather misleading to treat lack of a Materials statement as contributing to an index of irreproducibility. Certainly, there are many studies where this is the case, but it will vary from study to study depending on the nature of the research. Indeed, this may also be true for other components of the irreproducibility index: for instance, in a case study, there may be no analysis script because no statistical analysis was done. And in some papers, the raw data may all be present in the text already – that may be less common, but it is likely to be so for case studies, for instance.

A related point concerns the criteria for selecting papers for screening: it was surprising that the requirement for studies to have empirical data was not imposed at the outset: it should be possible to screen these out early on by specifying 'publication type'; instead, they were included and that means that the numbers used for the actual analysis are well below 400. The large number of non-empirical papers is not of particular relevance for the research questions considered here. In the Discussion, the authors expressed surprise at the large number of non-empirical papers they found; it would have been reasonable for them to depart from their preregistered plan on discovering this, and to review further papers to bring the number up to 400, restricting consideration to empirical papers only – also excluding case reports, which pose their own problems in this kind of analysis.

The analysis presented may create a backlash against metascientific analyses like this because it appears unfair on authors to use a metric based on criteria that may not apply to their study. If you are going to evaluate papers as to whether they include things like materials/data/ availability statements, then you need to have a N/A option. However, it may not be possible to rely on authors' self-evaluation of N/A and that means that metascientists doing an evaluation would need to read enough of the paper to judge whether such a statement should apply.

Some of the analyses could be dropped. The analysis of authorship by country, Figure 6, had too few cases for many countries to allow for sensible analysis.

It would be good to put into context efforts to replicate and reproduce papers and accessibility of materials, methods, data, code, etc – such as REPEAT initiative (https://www.repeatinitiative.org), the Reproducibility Project: Cancer Biology (https://elifesciences.org/collections/9b1e83d1/reproducibility-project-cancer-biology). Additionally, the discussion could benefit from emerging automated tools to assess transparency of materials, methods, data, code, etc (e.g., SciScore (https://www.jmir.org/2022/6/e37324), DataSeer (https://dataseer.ai/), and Ripeta (https://www.ncbi.nlm.nih.gov/pmc/articles/PMC8814593/)).

The abstract should specify that while 400 articles were screened, less than half were able to be assessed for the main findings reported – currently it is unclear.

The discussion could benefit from some concrete next steps. Such as what could be done at the journal level to make this more available? Items such as better metadata, incentivizing researchers (e.g., open science badges – https://journals.plos.org/plosbiology/article?id=10.1371/journal.pbio.1002456), reporting checklists (https://www.ncbi.nlm.nih.gov/pmc/articles/PMC5597130/), or checking articles for reproducibility (e.g., https://www.insidehighered.com/blogs/rethinking-research/should-journals-be-responsible-reproducibility) or replicability (e.g., http://www.orgsyn.org/). An overarching theme to expand on would be on what journals can do for improving their policies and incentive (e.g., https://www.science.org/doi/full/10.1126/science.aab2374). Similarly, this could be done at the funder and author level (and editor/reviewer).

Please consider reviewing the title as the authors study does not investigate reproducibility directly, but rather indirectly through a prerequisite of accessibility. So maybe consider reframing to have the title focused on the transparency 'audit' the authors did so readers do not think it should include a reproducibility 'audit'.

[Editors' note: further revisions were suggested prior to acceptance, as described below.]

Thank you for resubmitting your work entitled "Transparency of Research Practices in Cardiovascular Literature" for further consideration by *eLife*. Your revised article has been evaluated by a Reviewing Editor and a Senior Editor.

The manuscript has been improved but there are some remaining issues that need to be addressed, as outlined below:

(1) There is still a strong tendency to have the language center on replicability and reproducibility even though the article is entirely on transparency/accessibility of the aspects that would be needed for assessing replicability or reproducibility. While the title and abstract reflect this, it is not consistent throughout. For example, lines 277-282 still anchor on reproducible and replicable even though it's not tested – instead the authors are assessing if the materials needed for reproducibility and replicability are accessible from the published articles. Same with the figures (e.g., Figure 3).

(2) While the addition of additional projects is helpful for context, the authors might consider articles instead of websites. For example, REPEAT could be this article (https://www.nature.com/articles/s41467-022-32310-3). The reproducibility project in cancer biology could be this article (https://elifesciences.org/articles/67995) or this article (https://elifesciences.org/articles/71601). These would match the references to SciScore and Ripeta.

(3) Line 240 – it should be "Open Science Framework" not "Open Science Foundation".

(4) Lines 352-360 should have references instead of urls to the papers.

(5) In Table 3 – I think the authors should list any p-value over 1 as 1.0 – there is no additional value in reporting them as over 1.

(6) Figure 1F, 2A, 2B – at times the authors present data where the bar graph stops even though the number presented is less than the value (e.g., 1F goes to 564 for open access button even though the axis looks like it's less than 400). The authors seem to be adjusting their axis for clarity purposes, however, it is suggested the authors adjust further for increased clarity and decreased confusion – they should show the bar and axis as broken with the axis properly reflecting the value. Alternatively, the authors should create the graphs so they are reporting the entire range. Furthermore:

a) Figure 1 could more economically be shown in a single table.

b) If Figure 2 used stacked barplots rather than single bars, then it would be possible to show the relationship between study type and the dependent variables within the same plot – this would make it easier to gain an intuitive sense of what the chi square tests were showing

c) Figure 3 is unnecessary – the numbers reported could be more economically described in the text.

d) Figures 4 and 5 again miss an opportunity to show the interesting interactions between variables, by just plotting variables one at a time. For instance, Figure 4 panel A could use stacked bars to differentiate journals, which would make panel C unnecessary. Figure 5 could use stacked bars with study type denoted by colour, which would make it easier to economically show the 4 types of accessibility criteria in a single plot (ie one bar for each criterion, with the study types stacked in a bar). This website is v useful for showing how to do this in R: http://www.sthda.com/english/wiki/ggplot2-barplots-quick-start-guide-r-software-and-data-visualization

(7) There are issues with reproducibility. The file EmpiricalResearchOnly(3).csv could be open but It was not immediately clear how the Accessibility.Score.Fraction had been computed. Ideally, this should be done within the script. The criteria are defined in Table 1, but it is not easy to recompute this from the available script and file for two reasons: the responses are coded as text rather than numerical, and it is unclear how NA responses (which are very frequent for some items) are handled. When trying to recreate the Accessibility Score and the Accessibility Score proportion from the raw data, results were a little different (script appended below).

As far as this reviewer could see, the problem regarding cases where Materials may not be appropriate was only partially addressed. There is a code for Study.Type.With.No.Materials, but studies coded that way are also coded as "No" for Materials. Flag, and it seems that it is the latter that is still used to index the Repeatable/Reproducible variables?

It's possible this has all been addressed, but it is not at all clear. It would seem appropriate to ignore the Materials Flag when computing reproducibility etc. if the study type did not require materials – I checked the first study on file, and it was of this type. It was an analysis of mortality in data records, and was coded as Study.Type.With.No.Materials. As mentioned above, though, it isn't crystal clear how these computations were done and a revised script to automate the calculations within R is required, I think, for an article focused on reproducibility.

(8) The Fully.Reproducible and Fully.Repeatable variables are identical – and indeed that is how they are described on p. 4.

(9) Para 1 of Analysis says "We also screened whether studies could share Materials or not. For example, meta-analyses are empirical research studies, but do not typically have any shareable materials". Because the focus is now on empirical articles, it would be better to give an example of an empirical article that has no materials – the first study on the.csv file is of this type.

(10) Line 231 : should this be "across two levels"

(11) Line 266: "Only 14% of articles made their materials available (56 out of 393)" – need to break this down to show the proportion of papers after excluding those with no materials.

(12) Line 323; need comma or : rather than stop before "For example"

(13) Line 410: may be worth citing the work of Goldacre, showing that even when trials are pre-registered, variable switching is common when reporting results: Goldacre, B., et al. (2019). COMPare: A prospective cohort study correcting and monitoring 58 misreported trials in real time. Trials, 20(1), 118. https://doi.org/10.1186/s13063-019-3173-2

(14) Table 2 – it is not clear what the Yes vs No numbers are, or how these chi square values were computed. Could not find this analysis in the R script. what hypothesis is being tested here?

(15) Table 3 – because p-values greater than 1 make no sense, it may be better to follow the precedent of SPSS, and just censor these values so that the ceiling is 1. This could be explained in the legend.

Simple R script for trying to reproduce accessibility score

mydf <- read_csv(paste0(here("EmpiricalResearchOnly_3.csv")))

#csv file read from OSF: 'here' just ensures we look in working directory

#Use table 1 to create binary variables a1 to a17

mydf$a1 <- 1

mydf$a1[mydf$Study.Type.Clarity=="Inferred (needed to read)"]<-0 #no NA cases here

mydf$a2 <- 0

mydf$a2[mydf$Materials.Availability.Statement=="Yes the statement says that the materials (or some of the materials) are available."]<-1 #nb this turns NA to 0

mydf$a3<-0

mydf$a3[mydf$Data.Downloadable.Openable=="Yes"]<-1 #nb this turns NA to 0

mydf$a4<-1

mydf$a4[mydf$Data.Availability.Statement=="No – there is no data availability statement."]<-0

mydf$a5<-0

mydf$a5[mydf$Data.Downloadable.Openable=="Yes"]<-1 #nb this turns NA to 0

mydf$a6<-0

mydf$a6[mydf$Data.Clearly.Documented=="Yes"]<-1 #most here are NA

mydf$a7<-0

mydf$a7[mydf$Data.Contain.All.Raw.Data=="Yes"]<-1 #most here are NA

mydf$a8<-1

mydf$a8[mydf$Analysis.Script.Availability.Statement == "No – there is no analysis script availability statement."]<-0 #no NA here

mydf$a9<-0

mydf$a9[mydf$Analysis.Files.Downloadable.Openable=="Yes"]<-1 #many NA

mydf$a10<-0

mydf$a10[mydf$Pre.registered.Statement=="Yes – the statement says that there was a pre­-registration."]<-1 #no NA

mydf$a11=0

mydf$a11[mydf$Pre.registration.Accessible.Openable=="Yes"]<-1 #many NA

mydf$a12a=0

mydf$a12a[mydf$Pre.registration.Aspect…Hypothesis=="Hypotheses"]<-1

mydf$a12b=0

mydf$a12b[mydf$Pre.registration.Aspect…Methods=="Methods"]<-1

mydf$a12c=0

mydf$a12c[mydf$Pre.registration.Aspect…Methods=="Analysis.Plan"]<-1

mydf$a12<-mydf$a12a+mydf$a12b+mydf$a12c

mydf$a12[mydf$a12<3]<-0 #only code if all 3 are present

mydf$a12[mydf$a12==3]<-1

mydf$a13<-0

mydf$a13[mydf$Accessible.Protocol.Linked=="Yes"]<-1

mydf$a14a=0

mydf$a14a[mydf$Protocol.Aspect…Analysis.Plan=="Analysis plan"]<-1 #most are NA

mydf$a14b=0

mydf$a14b[mydf$Protocol.Aspect…Hypothesis=="Hypotheses"]<-1 #most are NA

mydf$a14c=0

mydf$a14c[mydf$Protocol.Aspect…Methods=="Methods"]<-1

mydf$a14<-mydf$a14a+mydf$a14b+mydf$a14c

mydf$a14[mydf$a14<3]<-0 #only code if all 3 are present

mydf$a14[mydf$a14==3]<-1

mydf$a15 <-1

mydf$a15[mydf$COI.Indicated=="No – there is no conflict of interest statement, and disclosures are NOT listed."]<-0 #no NA

mydf$a16<-0

myans<-substring(mydf$Funding.Sources.Statement,1,3)

w<-which(myans == "Yes")

mydf$a16[w]<-1

mydf$a17<-1

mydf$a17[mydf$Open.Access.Article=="No could not access article other than through paywall"]<-0

#no NA

mydf$allacc <-mydf$a1+mydf$a2+mydf$a3+mydf$a4+mydf$a5+mydf$a6+mydf$a7+mydf$a8+mydf$a9+mydf$a10+mydf$a11+mydf$a12+mydf$a13+mydf$a14+mydf$a15+mydf$a16+mydf$a17

mydf$accprop<-mydf$allacc/17

#Adjust for study type with no materials

w<-which(mydf$Study.Type.With.No.Materials=="Yes")

mydf$allacc[w] <- mydf$allacc[w]-mydf$a2[w]-mydf$a3[w] #remove items a2 and a3

mydf$accprop[w] <-mydf$allacc[w]/15 #adjust proportion

#visualise if computed accessibility score is same as Accessibility.Score (it isn't)

plot(mydf$allacc,mydf$Accessibility.Score)

abline(a=0,b=1)

plot(mydf$accprop,mydf$Accessibility.Score.Fraction)

abline(a=0,b=1)

[Editors' note: further revisions were suggested prior to acceptance, as described below.]

Thank you for resubmitting your work entitled "Transparency of Research Practices in Cardiovascular Literature" for further consideration by *eLife*. Your revised article has been evaluated by a Senior Editor and a Reviewing Editor.

The manuscript has been improved but there are some remaining issues that need to be addressed, as outlined below:

- Please reconsider the alternative visualizations of the figures. They don't have to remove their current figures, instead they can present alternative figure visualizations as supplementary figures so readers can see both versions.

- Figure legend 3, it should be "An article is considered 'partially reproducible' if any of data availability, analysis script…." (i.e., the 'if any' is missing).

- It is not sufficient to refer to an online calculator for the calculation of chi-square values. These are trivial to compute in R, and if they were part of the script, then it would be possible to work out which numbers had gone into the calculation. As it is, table 2 remains confusing because it shows numbers and a p-value for each of 4 categories of resource, but states that the chi-square test is used to test a 2-way relationship: between study type and resource. It seems we are shown only one dimension of the two-way table, which is not helpful. Furthermore, it is not clear where those numbers came from because it was unclear what variables were used.

- Given the paper's focus on reproducibility, there should be a script that generates all the tables in the paper. That allows readers to check on exactly what was done and is particularly useful if, for instance, an error is found at some point: the calculations can then be rerun to regenerate all tables and statistics (and ideally also the figures).

- Here's a bit of code that will at least do a chi-square test, and give the two-way data that is needed to interpret the relationship.

wantcat <- c('Clinical Case Study or Series','Clinical Observational Study','Clinical Trial','Laboratory Animal Study')

shortdf<-mydf[mydf$Study.Type %in% wantcat,] #filter Study.Type data

mytab <- table(shortdf$Study.Type,shortdf$Materials.Availability.Statement.Binary) #two way table

mychi <- chisq.test(mytab)

pcorr <- mychi$p.value*4 #multiply by 4 for Bonferroni-corrected value

#Display 2 way table and corrected pvalue for Table 2

mytab

pcorr

---

## [Author Response]

Essential Revisions (for the authors):The analysis states that to be fully reproducible, publications must include sufficient resources (materials, methods, data and analysis scripts). But how about cases where materials are not required to reproduce the work? In line 128-129 it is noted that the materials criterion was omitted for meta-analyses, but what about other types of study where materials may be either described adequately in the text, readily available (eg published questionnaires), or impossible to share (e.g. experimental animals). To see how valid these concerns might be, the first 4 papers were assessed in the deposited 'EmpricalResearchOnly.csv' file. Two had been coded as 'No Materials availability statement' and for two the value was blank.Study 1 used registry data and was coded as missing a Materials statement. The only materials that might be useful to have might be 'standardized case report forms' that were referred to. But the authors did note that the Registry methods were fully documented elsewhere.Study 2 was a short surgical case report – for this one the Materials field was left blank by the coder.Study 3 was a meta-analysis; the Materials field was blank by the coderStudy 4 was again coded as lacking a Material statement. It presented a model predicting outcome for cardiac arrhythmias. The definitions of the predictor variables were provided in supplementary materials. It is not clear what other materials might be needed.These four cases suggest that it is rather misleading to treat lack of a Materials statement as contributing to an index of irreproducibility. Certainly, there are many studies where this is the case, but it will vary from study to study depending on the nature of the research. Indeed, this may also be true for other components of the irreproducibility index: for instance, in a case study, there may be no analysis script because no statistical analysis was done. And in some papers, the raw data may all be present in the text already – that may be less common, but it is likely to be so for case studies, for instance.

We very much appreciate the reviewer’s point about how the calculated accessibility score may be inaccurate for articles that would not, by their nature, have materials, or, in some cases, analyses. Our screening protocol was directly adapted from Iqbal et al. (2016) with very few changes. The reviewer requested clarification on how we decided a study would be expected or not expected to be able to make materials available. We determined whether articles could have material availability based on their study type and by interpretation of the abstract. For example, meta analysis were a study type that was never expected to have materials available. We did make the assumption that all empirical research articles would include methods, data, and some form of analysis. We have now updated the text to more clearly explain our definition so the readers will understand how the data was generated (see page 3). We have also included an additional paragraph in the Limitations section (see page 10) to better explain how the accessibility score derived in this study is an imperfect indicator for some studies.

A related point concerns the criteria for selecting papers for screening: it was surprising that the requirement for studies to have empirical data was not imposed at the outset: it should be possible to screen these out early on by specifying 'publication type'; instead, they were included and that means that the numbers used for the actual analysis are well below 400. The large number of non-empirical papers is not of particular relevance for the research questions considered here. In the Discussion, the authors expressed surprise at the large number of non-empirical papers they found; it would have been reasonable for them to depart from their preregistered plan on discovering this, and to review further papers to bring the number up to 400, restricting consideration to empirical papers only – also excluding case reports, which pose their own problems in this kind of analysis.

We appreciate the reviewer’s perspective regarding our effective sample size for empirical research papers and we agree that it is reasonable to depart from our pre-registered plan upon analysis of a large number of non-empirical papers. An additional 239 papers were screened to bring the total number of empirical papers to 393. As an introductory paper describing a developing accessibility score in cardiovascular literature, we also included some data regarding the number of non-empirical studies in a random selection of publications, and their funding and conflict of interest information.

The analysis presented may create a backlash against metascientific analyses like this because it appears unfair on authors to use a metric based on criteria that may not apply to their study. If you are going to evaluate papers as to whether they include things like materials/data/ availability statements, then you need to have a N/A option. However, it may not be possible to rely on authors' self-evaluation of N/A and that means that metascientists doing an evaluation would need to read enough of the paper to judge whether such a statement should apply.

Our coding form allowed a ‘N/A’ option for the inclusion of materials for empirical research studies, but our study did make the assumption that all empirical research would include a shareable protocol, data, and analysis. We agree that this assumption may not be valid for a small proportion of articles and we have expanded our discussion to more thoroughly describe this limitation. All screeners were prepared to read enough of the paper to judge whether the selection of ‘N/A’ for ‘Materials availability’ could apply.

Some of the analyses could be dropped. The analysis of authorship by country, Figure 6, had too few cases for many countries to allow for sensible analysis.

We appreciate and agree with the reviewer’s concern. We have now removed figure six and any discussion or analysis of the author country of origin. We have now updated the Methods section to explain why we are not including this analysis.

It would be good to put into context efforts to replicate and reproduce papers and accessibility of materials, methods, data, code, etc – such as REPEAT initiative (https://www.repeatinitiative.org), the Reproducibility Project: Cancer Biology (https://elifesciences.org/collections/9b1e83d1/reproducibility-project-cancer-biology). Additionally, the discussion could benefit from emerging automated tools to assess transparency of materials, methods, data, code, etc (e.g., SciScore (https://www.jmir.org/2022/6/e37324), DataSeer (https://dataseer.ai/), and Ripeta (https://www.ncbi.nlm.nih.gov/pmc/articles/PMC8814593/)).

We appreciate the reviewer’s making us aware of these studies and tools. We have now expanded the introduction to describe current efforts to replicate and reproduce papers and emerging tools to assess transparency (page 2.)

The abstract should specify that while 400 articles were screened, less than half were able to be assessed for the main findings reported – currently it is unclear.

We have now updated the abstract to more clearly explain the number of articles screened. We also increased the number of empirical research articles screened to 393 (page 1).

The discussion could benefit from some concrete next steps. Such as what could be done at the journal level to make this more available? Items such as better metadata, incentivizing researchers (e.g., open science badges – https://journals.plos.org/plosbiology/article?id=10.1371/journal.pbio.1002456), reporting checklists (https://www.ncbi.nlm.nih.gov/pmc/articles/PMC5597130/), or checking articles for reproducibility (e.g., https://www.insidehighered.com/blogs/rethinking-research/should-journals-be-responsible-reproducibility) or replicability (e.g., http://www.orgsyn.org/). An overarching theme to expand on would be on what journals can do for improving their policies and incentive (e.g., https://www.science.org/doi/full/10.1126/science.aab2374). Similarly, this could be done at the funder and author level (and editor/reviewer).

We completely agree and we have now expanded the Discussion section to emphasize the steps that journals and funders can take to improve transparent research practices (page 9).

Please consider reviewing the title as the authors study does not investigate reproducibility directly, but rather indirectly through a prerequisite of accessibility. So maybe consider reframing to have the title focused on the transparency 'audit' the authors did so readers do not think it should include a reproducibility 'audit'.

We have changed the title from “Accessibility and Reproducible Research Practices in Cardiovascular Literature” to “Transparency of Research Practices in Cardiovascular Literature.” (page 1).

[Editors’ note: what follows is the authors’ response to the second round of review.]

The manuscript has been improved but there are some remaining issues that need to be addressed, as outlined below:(1) There is still a strong tendency to have the language center on replicability and reproducibility even though the article is entirely on transparency/accessibility of the aspects that would be needed for assessing replicability or reproducibility. While the title and abstract reflect this, it is not consistent throughout. For example, lines 277-282 still anchor on reproducible and replicable even though it's not tested – instead the authors are assessing if the materials needed for reproducibility and replicability are accessible from the published articles.

We have changed the language in the introduction to clarify that our study provides information about the availability of materials needed to facilitate reproduction/replication of a research study, not the reproducibility/replication study itself (page 3).

Same with the figures (e.g., Figure 3).

We have similarly clarified the language in the figure 3 caption (page 17).

(2) While the addition of additional projects is helpful for context, the authors might consider articles instead of websites. For example, REPEAT could be this article (https://www.nature.com/articles/s41467-022-32310-3). The reproducibility project in cancer biology could be this article (https://elifesciences.org/articles/67995) or this article (https://elifesciences.org/articles/71601). These would match the references to SciScore and Ripeta.

We thank the reviewers for additional references. We have expanded the introduction to incorporate all suggested references to match to SciScore, Ripeta, and REPEAT (page 2).

(3) Line 240 – it should be "Open Science Framework" not "Open Science Foundation".

We changed “Open Science Foundation” to “Open Science Framework” (page 6).

(4) Lines 352-360 should have references instead of urls to the papers.

We have converted all urls into references, and added new references to the citations (page 8).

(5) In Table 3 – I think the authors should list any p-value over 1 as 1.0 – there is no additional value in reporting them as over 1.

We have updated the p-values in table 3 to have a ceiling of 1 (page 13).

(6) Figure 1F, 2A, 2B – at times the authors present data where the bar graph stops even though the number presented is less than the value (e.g., 1F goes to 564 for open access button even though the axis looks like it's less than 400). The authors seem to be adjusting their axis for clarity purposes, however, it is suggested the authors adjust further for increased clarity and decreased confusion – they should show the bar and axis as broken with the axis properly reflecting the value. Alternatively, the authors should create the graphs so they are reporting the entire range. Furthermore:a) Figure 1 could more economically be shown in a single table.

We have now fixed the axes for figure 1. Although we agree that this data could also be shown in a table, we prefer to display the information with a figure to help with readability. Additionally, because each element (Panels A-F) has different variables, we would need to use a similar arrangement of six subtables.

b) If Figure 2 used stacked barplots rather than single bars, then it would be possible to show the relationship between study type and the dependent variables within the same plot – this would make it easier to gain an intuitive sense of what the chi square tests were showing

We appreciate the reviewer’s observation and we have updated the figure axes for accuracy. Although stacked barplots may help show relationships across variables, we prefer to show individual bars to ensure readability.

c) Figure 3 is unnecessary – the numbers reported could be more economically described in the text.

Although we report the numbers for figure 3 in the text, we prefer to also show the data graphically to help with readability.

d) Figures 4 and 5 again miss an opportunity to show the interesting interactions between variables, by just plotting variables one at a time. For instance, Figure 4 panel A could use stacked bars to differentiate journals, which would make panel C unnecessary. Figure 5 could use stacked bars with study type denoted by colour, which would make it easier to economically show the 4 types of accessibility criteria in a single plot (ie one bar for each criterion, with the study types stacked in a bar). This website is v useful for showing how to do this in R: http://www.sthda.com/english/wiki/ggplot2-barplots-quick-start-guide-r-software-and-data-visualization

We again appreciate the reviewer’s suggestion to use stacked bars, however we prefer to use single bars to help with readability. It is very hard for readers to compare values across stacked bars because of the lack of axis references for all but the outer- and inner-most bar segments.

(7) There are issues with reproducibility. The file EmpiricalResearchOnly(3).csv could be open but It was not immediately clear how the Accessibility.Score.Fraction had been computed. Ideally, this should be done within the script. The criteria are defined in Table 1, but it is not easy to recompute this from the available script and file for two reasons: the responses are coded as text rather than numerical, and it is unclear how NA responses (which are very frequent for some items) are handled. When trying to recreate the Accessibility Score and the Accessibility Score proportion from the raw data, results were a little different (script appended below).As far as this reviewer could see, the problem regarding cases where Materials may not be appropriate was only partially addressed. There is a code for Study.Type.With.No.Materials, but studies coded that way are also coded as "No" for Materials. Flag, and it seems that it is the latter that is still used to index the Repeatable/Reproducible variables?It's possible this has all been addressed, but it is not at all clear. It would seem appropriate to ignore the Materials Flag when computing reproducibility etc. if the study type did not require materials – I checked the first study on file, and it was of this type. It was an analysis of mortality in data records, and was coded as Study.Type.With.No.Materials. As mentioned above, though, it isn't crystal clear how these computations were done and a revised script to automate the calculations within R is required, I think, for an article focused on reproducibility.

We thank the reviewer for their insights and for taking the time to develop an R script to calculate the Accessibility Score. We had previously calculated this score using Tableau, but we agree it would be more transparent and reproducible to calculate this variable using a script. We have therefore created an R script to calculate the Accessibility Score and made this code available on OSF.

Creating this code did reveal a discrepancy in our Tableau-calculation for the accessibility score, and our current calculation now aligns with that generated by the code provided by the reviewer. We have updated the figures and manuscript to reflect the changed data.

(8) The Fully.Reproducible and Fully.Repeatable variables are identical – and indeed that is how they are described on p. 4.

We have corrected the statement to reflect our original screening criteria and definitions of reproducible and repeatability (page 4).

(9) Para 1 of Analysis says "We also screened whether studies could share Materials or not. For example, meta-analyses are empirical research studies, but do not typically have any shareable materials". Because the focus is now on empirical articles, it would be better to give an example of an empirical article that has no materials – the first study on the.csv file is of this type.

We have included an example of an empirical article that has no materials (page 4) and included a citation (page 19).

(10) Line 231 : should this be "across two levels"

We have corrected this typo (page 5).

(11) Line 266: "Only 14% of articles made their materials available (56 out of 393)" – need to break this down to show the proportion of papers after excluding those with no materials.

We made an additional clarification statement that the 393 papers mentioned in this statement have theoretically available materials, meaning these articles do not include papers with no materials (page 6). This statement is also made in the caption of Figure 2B.

(12) Line 323; need comma or : rather than stop before "For example"

We have added a comma (page 7).

(13) Line 410: may be worth citing the work of Goldacre, showing that even when trials are pre-registered, variable switching is common when reporting results: Goldacre, B., et al. (2019). COMPare: A prospective cohort study correcting and monitoring 58 misreported trials in real time. Trials, 20(1), 118. https://doi.org/10.1186/s13063-019-3173-2

We agree and now reference this study in the Discussion (page 9).

(14) Table 2 – it is not clear what the Yes vs No numbers are, or how these chi square values were computed. Could not find this analysis in the R script. what hypothesis is being tested here?

We are testing the hypothesis that there is a relationship between study type and the presence of materials, methods, data, and analysis. We added this to our table caption and clarified that “yes” refers to the number of articles that have the resource and “no” refers to the number of articles that do not have the resource. We used an online calculator, not an R script to calculate the scores. We have updated the methods and references to cite this tool (page 13).

(15) Table 3 – because p-values greater than 1 make no sense, it may be better to follow the precedent of SPSS, and just censor these values so that the ceiling is 1. This could be explained in the legend.

We followed the precedent of SPSS as suggested and explained why values over 1 were censored in the caption (page 13).

[Editors’ note: what follows is the authors’ response to the third round of review.]

The manuscript has been improved but there are some remaining issues that need to be addressed, as outlined below:- Please reconsider the alternative visualizations of the figures. They don't have to remove their current figures, instead they can present alternative figure visualizations as supplementary figures so readers can see both versions.

We appreciate the reviewer’s suggestion, but we respectfully decline remaking alternative visualizations of the figures. Because supplementary figures are typically used to show additional data, as opposed to alternative visualizations or data that is already shown, we think that additional visualizations of the same data may confuse the readers.

- Figure legend 3, it should be "An article is considered 'partially reproducible' if any of data availability, analysis script…." (i.e., the 'if any' is missing).

Figure legend 3 was corrected to include “if any of.” (page 17)

- It is not sufficient to refer to an online calculator for the calculation of chi-square values. These are trivial to compute in R, and if they were part of the script, then it would be possible to work out which numbers had gone into the calculation. As it is, table 2 remains confusing because it shows numbers and a p-value for each of 4 categories of resource, but states that the chi-square test is used to test a 2-way relationship: between study type and resource. It seems we are shown only one dimension of the two-way table, which is not helpful. Furthermore, it is not clear where those numbers came from because it was unclear what variables were used.

We sincerely appreciate the reviewer’s concerns regarding the reproducibility of the statistical analyses. We now provide R code to accompany the paper that calculate the chi-square values used in Table 2 and Table 3. Note that upon recalculating these tests in R, we obtained slightly different values than in the previous manuscript submission; however none of the implications have changed. We have updated the methods (page 5), and the tables themselves (p13) accordingly.

- Given the paper's focus on reproducibility, there should be a script that generates all the tables in the paper. That allows readers to check on exactly what was done and is particularly useful if, for instance, an error is found at some point: the calculations can then be rerun to regenerate all tables and statistics (and ideally also the figures).- Here's a bit of code that will at least do a chi-square test, and give the two-way data that is needed to interpret the relationship.wantcat <- c('Clinical Case Study or Series','Clinical Observational Study','Clinical Trial','Laboratory Animal Study')shortdf<-mydf[mydf$Study.Type %in% wantcat,] #filter Study.Type datamytab <- table(shortdf$Study.Type,shortdf$Materials.Availability.Statement.Binary) #two way tablemychi <- chisq.test(mytab)pcorr <- mychi$p.value*4 #multiply by 4 for Bonferroni-corrected value#Display 2 way table and corrected pvalue for Table 2mytabpcorr

We appreciate the reviewer’s perspective regarding the reproducibility of the analysis methods. We now include R code that calculates the values for tables 2 and 3, as well as the associated chi-square tests. Table 1 and Table 4 are not generated from the original data and cannot be generated in R.